# RETHINKING THE HYPERPARAMETERS FOR FINE-TUNING

**Hao Li[1], Pratik Chaudhari[2]\*, Hao Yang[1], Michael Lam[1], Avinash Ravichandran[1], Rahul Bhotika[1], Stefano Soatto[1,3]**

[1]Amazon Web Services, [2]University of Pennsylvania, [3]University of California, Los Angeles

{haolimax, haoyng, michlam, ravinash, bhotikar, soattos}@amazon.com, pratikac@seas.upenn.edu

## ABSTRACT

Fine-tuning from pre-trained ImageNet models has become the de-facto standard for various computer vision tasks. Current practices for fine-tuning typically involve selecting an ad-hoc choice of hyperparameters and keeping them fixed to values normally used for training from scratch. This paper re-examines several common practices of setting hyperparameters for fine-tuning. Our findings are based on extensive empirical evaluation for fine-tuning on various transfer learning benchmarks. (1) While prior works have thoroughly investigated learning rate and batch size, momentum for fine-tuning is a relatively unexplored parameter. We find that the value of momentum also affects fine-tuning performance and connect it with previous theoretical findings. (2) Optimal hyperparameters for fine-tuning, in particular, the effective learning rate, are not only dataset dependent but also sensitive to the similarity between the source domain and target domain. This is in contrast to hyperparameters for training from scratch. (3) Reference-based regularization that keeps models close to the initial model does not necessarily apply for "dissimilar" datasets. Our findings challenge common practices of fine-tuning and encourages deep learning practitioners to rethink the hyperparameters for fine-tuning.

## 1 INTRODUCTION

Many real-world applications often have a limited number of training instances, which makes directly training deep neural networks hard and prone to overfitting. Transfer learning with the knowledge of models learned on a similar task can help to avoid overfitting. Fine-tuning is a simple and effective approach of transfer learning and has become popular for solving new tasks in which pre-trained models are fine-tuned with the target dataset. Specifically, fine-tuning on pre-trained ImageNet classification models (Simonyan & Zisserman, 2015; He et al., 2016b) has achieved impressive results for tasks such as object detection (Ren et al., 2015) and segmentation (He et al., 2017; Chen et al., 2017) and is becoming the de-facto standard of solving computer vision problems. It is believed that the weights learned on the source dataset with a large number of instances provide better initialization for the target task than random initialization. Even when there is enough training data, fine-tuning is still preferred as it often reduces training time significantly (He et al., 2019).

The common practice of fine-tuning is to adopt the default hyperparameters for training large models while using smaller initial learning rate and shorter learning rate schedule. It is believed that adhering to the original hyperparameters for fine-tuning with small learning rate prevents destroying the originally learned knowledge or features. For instance, many studies conduct fine-tuning of ResNets (He et al., 2016b) with these default hyperparameters: learning rate 0.01, momentum 0.9 and weight decay 0.0001. However, the default setting is not necessarily optimal for fine-tuning on other tasks. While few studies have performed extensive hyperparameter search for learning rate and weight decay (Mahajan et al., 2018; Kornblith et al., 2019), the momentum coefficient is rarely changed. Though the effectiveness of the hyperparameters has been studied extensively for training a model from scratch, how to set the hyperparameters for fine-tuning is not yet fully understood.

---

\*Work done while at Amazon Web Services

In addition to using ad-hoc hyperparameters, commonly held beliefs for fine-tuning also include:

- Fine-tuning pre-trained networks outperforms training from scratch; recent work (He et al., 2019) has already revisited this.
- Fine-tuning from similar domains and tasks works better (Ge & Yu, 2017; Cui et al., 2018; Achille et al., 2019; Ngiam et al., 2018).
- Explicit regularization with initial models matters for transfer learning performance (Li et al., 2018; 2019).

Are these practices or beliefs always valid? From an optimization perspective, the difference between fine-tuning and training from scratch is all about the initialization. However, the loss landscape of the pre-trained model and the fine-tuned solution could be much different, so as their optimization strategies and hyperparameters. Would the hyperparameters for training from scratch still be useful for fine-tuning? In addition, most of the hyperparameters (e.g., batch size, momentum, weight decay) are frozen; will the conclusion differ when some of them are changed?

With these questions in mind, we re-examined the common practices for fine-tuning. We conducted extensive hyperparameter search for fine-tuning on various transfer learning benchmarks with different source models. The goal of our work is not to obtain state-of-the-art performance on each fine-tuning task, but to understand the effectiveness of each hyperparameter for fine-tuning, avoiding unnecessary computation. We explain why certain hyperparameters work so well on certain datasets while fail on others, which can guide hyperparameter search for fine-tuning.

Our main findings are as follows:

- Optimal hyperparameters for fine-tuning are not only dataset dependent, but are also dependent on the similarity between the source and target domains, which is different from training from scratch. Therefore, the common practice of using optimization schedules derived from ImageNet training cannot guarantee good performance. It explains why some tasks are not achieving satisfactory results after fine-tuning because of inappropriate hyperparameter selection. Specifically, as opposed to the common practice of rarely tuning the momentum value beyond 0.9, we find that zero momentum sometimes work better for fine-tuning on tasks that are similar with the source domain, while nonzero momentum works better for target domains that are different from the source domain.
- Hyperparameters are coupled together and it is the effective learning rate—which encapsulates the learning rate and momentum—that matters for fine-tuning performance. While effective learning rate has been studied for training from scratch, to the best of our knowledge, no previous work investigates effective learning rate for fine-tuning and is less used in practice. Our observation of momentum can be explained as small momentum actually decreases the effective learning rate, which is more suitable for fine-tuning on similar tasks. We show that the optimal effective learning rate depends on the similarity between the source and target domains.
- We find regularization methods that were designed to keep models close to the initial model does not necessarily work for "dissimilar" datasets, especially for nets with Batch Normalization. Simple weight decay can result in as good performance as the reference-based regularization methods for fine-tuning with better search space.

## 2 RELATED WORK

In transfer learning for image classification, the last layer of a pre-trained network is usually replaced with a randomly initialized fully connected layer with the same size as the number of classes in the target task (Simonyan & Zisserman, 2015). It has been shown that fine-tuning the whole network usually results in better performance than using the network as a static feature extractor (Yosinski et al., 2014; Donahue et al., 2014; Huh et al., 2016; Mormont et al., 2018; Kornblith et al., 2019). Ge & Yu (2017) select images that have similar local features from source domain to jointly fine-tune pre-trained networks. Cui et al. (2018) estimate domain similarity with ImageNet and demonstrate that transfer learning benefits from pre-training on a similar source domain. Besides image classification, many object detection frameworks also rely on fine-tuning to improve over training from scratch (Girshick et al., 2014; Ren et al., 2015).

Many researchers re-examined whether fine-tuning is a necessity for obtaining good performance. Ngiam et al. (2018) find that when domains are mismatched, the effectiveness of transfer learning is negative, even when domains are intuitively similar. Kornblith et al. (2019) examine the fine-tuning performance of various ImageNet models and find a strong correlation between ImageNet top-1 accuracy and the transfer accuracy. They also find that pre-training on ImageNet provides minimal benefits for some fine-grained object classification dataset. He et al. (2019) questioned whether ImageNet pre-training is necessary for training object detectors. They find the solution of training from scratch is no worse than the fine-tuning counterpart as long as the target dataset is large enough. Raghu et al. (2019) find that transfer learning has negligible performance boost on medical imaging applications, but speed up the convergence significantly.

There are many literatures on hyperparameter selection for training neural networks from scratch, mostly on batch size, learning rate and weight decay (Goyal et al., 2017; Smith et al., 2018; Smith & Topin, 2019). There are few works on the selection of momentum (Sutskever et al., 2013). Zhang & Mitliagkas (2017) proposed an automatic tuner for momentum and learning rate in SGD. There are also studies on the correlations of the hyperparameters, such as linear scaling rule between batch size and learning (Goyal et al., 2017; Smith et al., 2018; Smith, 2017). However, most of these advances on hyperparameter tuning are designed for training from scratch and have not examined on fine-tuning tasks for computer vision problems. Most work on fine-tuning simply choose fixed hyperparameters (Cui et al., 2018) or use dataset-dependent learning rates (Li et al., 2018) in their experiments. Due to the huge computational cost for hyperparameter search, only a few works (Kornblith et al., 2019; Mahajan et al., 2018) performed large-scale grid search of learning rate and weight decay for obtaining the best performance.

## 3  TUNING HYPERPARAMETERS FOR FINE-TUNING

In this section, we first introduce the notations and experimental settings, and then present our observations on momentum, effective learning rate and regularization. The fine-tuning process is not different from learning from scratch except for the weights initialization. The goal of the process is still to minimize the objective function $L = \frac{1}{N} \sum_{i=1}^{N} \ell(f(x_i, \theta), y_i) + \frac{\lambda}{2} \|\theta\|_2^2$, where $\ell$ is the loss function, $N$ is the number of samples, $x_i$ is the input data, $y_i$ is its label, $f$ is the neural network, $\theta$ is the model parameters and $\lambda$ is the regularization hyperparameter or weight decay. Momentum is widely used for accelerating and smoothing the convergence of SGD by accumulating a velocity vector in the direction of persistent loss reduction (Polyak, 1964; Sutskever et al., 2013; Goh, 2017). The commonly used Nesterov's Accelerated Gradient (Nesterov, 1983) is given by:

$$v_{t+1} = mv_t - \eta_t \frac{1}{n} \sum_{i=1}^{n} \nabla \ell(f(x_i, \theta_t + mv_t), y_i) \tag{1}$$

$$\theta_{t+1} = \theta_t + v_{t+1} - \eta\lambda\theta_t \tag{2}$$

where $\theta_t$ indicates the model parameters at iteration $t$. The hyperparameters include the learning rate $\eta_t$, batch size $n$, momentum coefficient $m \in [0, 1)$, and the weight decay $\lambda$.

### 3.1  EXPERIMENTAL SETTINGS

We evaluate fine-tuning on seven widely used image classification datasets, which covers tasks for fine-grained object recognition, scene recognition and general object recognition. Detailed statistics of each dataset can be seen in Table 1. We use ImageNet (Russakovsky et al., 2015), Places-365 (Zhou et al., 2018) and iNaturalist (Van Horn et al., 2018) as source domains for pre-trained models. We resize the input images such that the aspect ratio is preserved and the shorter side is 256 pixels. The images are normalized with mean and std values calculated over ImageNet. For data augmentation, we adopt the common practices used for training ImageNet models (Szegedy et al., 2015): random mirror, random scaled cropping with scale and aspect variations, and color jittering. The augmented images are resized to 224×224. Note that state-of-the-art results could achieve even better performance by using higher resolution images (Cui et al., 2018) or better data augmentation (Cubuk et al., 2018).

We mainly use ResNet-101-V2 (He et al., 2016a) as our base network, which is pre-trained on ImageNet (Russakovsky et al., 2015). Similar observations are also made on DenseNets (Huang et al., 2017) and MobileNet (Howard et al., 2017). The hyperparameters to be tuned (and ranges)

Table 1: Datasets statistics. For the Caltech-256 dataset, we randomly sampled 60 images for each class following the procedure used in (Li et al., 2018). For the Aircraft and Flower dataset, we combined the original training set and validation set and evaluated on the test set. For iNat 2017, we combined the original training set and 90% of the validation set following (Cui et al., 2018).

| Datasets | Task Category | Classes | Training | Test |
|---|---|---|---|---|
| Oxford Flowers (Nilsback & Zisserman, 2008) | fine-grained object recog. | 102 | 2,040 | 6,149 |
| CUB-Birds 200-2011 (Wah et al., 2011) | fine-grained object recog. | 200 | 5,994 | 5,794 |
| FGVC Aircrafts (Maji et al., 2013) | fine-grained object recog. | 100 | 6,667 | 3,333 |
| Stanford Cars (Krause et al., 2013) | fine-grained object recog. | 196 | 8,144 | 8,041 |
| Stanford Dogs (Khosla et al., 2011) | fine-grained object recog. | 120 | 12,000 | 8,580 |
| MIT Indoor-67 (Sharif Razavian et al., 2014) | scene classification | 67 | 5,360 | 1,340 |
| Caltech-256-60 (Griffin et al., 2007) | general object recog. | 256 | 15,360 | 15,189 |
| iNaturalist 2017 (Van Horn et al., 2018) | fine-grained object recog. | 5,089 | 665,571 | 9,599 |
| Place365 (Zhou et al., 2018) | scene classification | 365 | 1,803,460 | 36,500 |

are: learning rate (0.1, 0.05, 0.01, 0.005, 0.001, 0.0001), momentum (0.9, 0.99, 0.95, 0.9, 0.8, 0.0) and weight decay (0.0, 0.0001, 0.0005, 0.001). We set the *default* hyperparameters to be batch size 256[1], learning rate 0.01, momentum 0.9 and weight decay 0.0001. To avoid insufficient training and observe the complete convergence behavior, we use 300 epochs for fine-tuning and 600 epochs for scratch-training, which is long enough for the training curves to converge. The learning rate is decayed by a factor of 0.1 at epoch 150 and 250. We report the Top-1 validation (test) error at the end of training. The total computation time for the experiments is more than 10K GPU hours.

## 3.2 EFFECT OF MOMENTUM AND DOMAIN SIMILARITY

Momentum 0.9 is the most widely used value for training from scratch (Krizhevsky et al., 2012; Simonyan & Zisserman, 2015; He et al., 2016b) and is also widely adopted for fine-tuning (Kornblith et al., 2019). To the best of our knowledge, it is rarely changed, regardless of the network architectures or target tasks. To check the influence of momentum on fine-tuning, we first search for the best momentum value for fine-tuning on the Birds dataset with different weight decay and learning rate. Figure 1(a) shows the performance of fine-tuning with and without weight decays. Surprisingly, momentum zero actually outperforms the nonzero momentum. The optimal learning rate also increases when the momentum is disabled as shown in Figure 1(b).

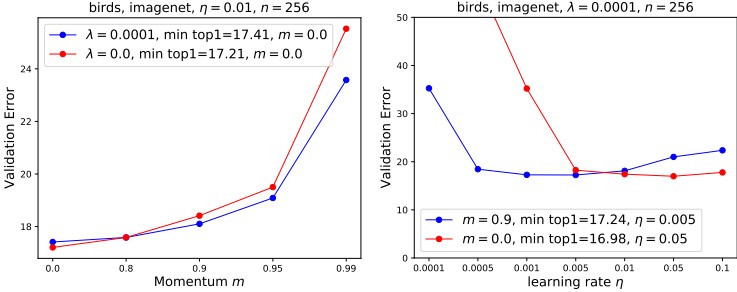

Figure 1: (a) Searching for the optimal momentum on Birds dataset with fixed learning rate 0.01 and different weight decays. Detailed learning curves and results of other hyperparameters can be found in Appendix A. (b) Comparison of momentum 0.9 and 0.0 with different learning rates on the Birds dataset, $\lambda$ is fixed at 0.0001.

To verify this observation, we further compare momentum 0.9 and 0.0 on other datasets. Table 2 shows the performance of 8 hyperparameter settings on 7 datasets. We observe a clear pattern that disabling momentum works better for Dogs, Caltech and Indoor, while momentum 0.9 works better for Cars, Aircrafts and Flowers.

---

[1] For each training job with ResNet-101 and batch size 256, we use 8 NVIDIA Tesla V100 GPUs for synchronous training, where each GPU uses a batch of 32 and no SyncBN is used.

Table 2: Top-1 validation errors on seven datasets by fine-tuning ImageNet pre-trained ResNet-101 with different hyperparmeters. Each row represents a network fine-tuned by a set of hyperparameters (left four columns). The datasets are ranked by the relative improvement by disabling momentum. The lowest error rates with the same momentum are marked as bold. Note that the performance difference for Birds is not very significant.

| $m$ | $\eta$ | $\lambda$ | Dogs | Caltech | Indoor | Birds | Cars | Aircrafts | Flowers |
|---|---|---|---|---|---|---|---|---|---|
| 0.9 | 0.01 | 0.0001 | 17.20 | 14.85 | 23.76 | 18.10 | 9.10 | 17.55 | **3.12** |
| 0.9 | 0.01 | 0 | 17.41 | 14.51 | 24.59 | 18.42 | 9.60 | **17.40** | 3.33 |
| 0.9 | 0.005 | 0.0001 | 14.14 | 13.42 | 24.59 | 17.24 | **9.08** | 18.21 | 3.50 |
| 0.9 | 0.005 | 0 | 14.80 | 13.67 | 22.79 | 17.54 | 9.31 | 17.82 | 3.53 |
| 0 | 0.01 | 0.0001 | 11.00 | 12.11 | 21.14 | 17.41 | 11.07 | 20.58 | 5.48 |
| 0 | 0.01 | 0 | 10.87 | 12.16 | 21.29 | **17.21** | 10.65 | 20.46 | 5.25 |
| 0 | 0.005 | 0.0001 | 10.21 | 11.86 | 21.96 | 18.24 | 13.22 | 24.39 | 7.03 |
| 0 | 0.005 | 0 | **10.12** | **11.61** | **20.76** | 18.40 | 13.11 | 23.91 | 6.78 |

Table 3: Verification of the effect of momentum on other source domains rather than ImageNet. The hyperparameters are $n = 256$, $\eta = 0.01$, and $\lambda = 0.0001$. Momentum 0 works better for transferring from iNat-2017 to Birds and transferring from Places-365 to Indoor comparing to momentum 0.9 counterparts.

| Source domain | $m$ | Indoor | Birds | Dogs | Caltech | Cars | Aircrafts |
|---|---|---|---|---|---|---|---|
| iNat-2017 | 0.9 | **30.73** | 14.69 | 24.74 | **20.12** | **11.16** | **19.86** |
| | 0 | 34.11 | **12.29** | **23.87** | 21.47 | 16.89 | 27.21 |
| Places-365 | 0.9 | 22.19 | **27.72** | 30.84 | 22.53 | **11.06** | **21.27** |
| | 0 | **20.16** | 32.17 | 32.47 | 22.60 | 14.67 | 25.29 |

Interestingly, datasets such as Dogs, Caltech, Indoor and Birds are known to have high overlap with ImageNet dataset[2], while Cars and Aircrafts are identified to be difficult to benefit from fine-tuning from pre-trained ImageNet models (Kornblith et al., 2019). According to Cui et al. (2018), in which the Earth Mover's Distance (EMD) is used to calculate the similarity between ImageNet and other domains, the similarity to Dogs and Birds are 0.619 and 0.563, while the similarity to Cars, Aircrafts and Flowers are 0.560, 0.556 and 0.525[3]. The relative order of similarities to ImageNet is

*Dogs, Birds, Cars, Aircrafts and Flowers*

which aligns well with the transition of optimal momentum value from 0.0 to 0.9. Following the similarity calculation, we can also verified Caltech and Indoor are more close to ImageNet than Cars/Aircrafts/Flowers (Table 3.3).

To verify the connection between momentum and domain similarity, we further fine-tune from different source domains such as Places-365 and iNaturalist, which are known to be better source domains than ImageNet for fine-tuning on Indoor and Birds dataset (Cui et al., 2018). We may expect that fine-tuning from iNaturalist works better for Birds with $m = 0$ and similarly, Places for Indoor. Indeed, as shown in Table 3, disabling momentum improves the performance when the source and target domain are similar, such as Places for Indoor and iNaturalist for Birds.

**Small momentum works better for fine-tuning on domains that are close to the source domain**
One explanation for the above observations is that because the Dogs dataset is very close to ImageNet, the pre-trained ImageNet model is expected to be close to the fine-tuned solution on the Dogs dataset. In this case, momentum may not help much as the gradient direction around the minimum could be much random and accumulating the momentum direction could be meaningless. Whereas, for

---

[2]Stanford Dogs (Khosla et al., 2011) was built using images and annotation from ImageNet for the task of fine-grained image categorization. Caltech-256 has at least 200 categories exist in ImageNet (Deng et al., 2010). Images in the CUB-Birds dataset overlap with images in ImageNet.

[3]The domain similarity caluculation is discussed in Appendix B and the exact value can be found in Table 3.3.

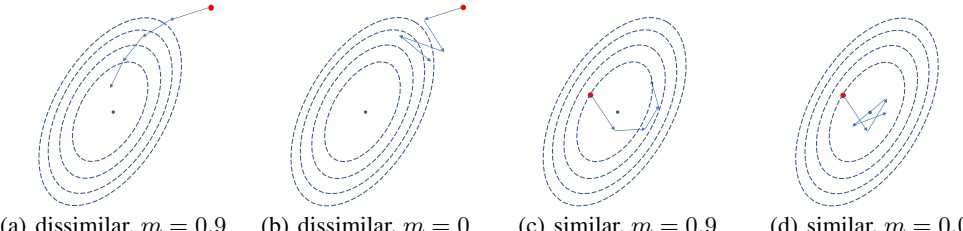

(a) dissimilar, $m = 0.9$    (b) dissimilar, $m = 0$    (c) similar, $m = 0.9$    (d) similar, $m = 0.0$

Figure 2: An illustration of the effect of momentum on different fine-tuning scenarios from the loss-landscape perspective. The red point is the pre-trained model and the blue point is the fine-tuned solution. The dashed lines are loss contours. Assuming the step size is fixed, large momentum accelerates the convergence when the initialization is far from the minimum ((a) and (b)). On the contrary, large momentum may impede the convergence as shown in (c) and (d) when the initialization is close to the minimum.

faraway target domains (e.g., Cars and Aircrafts) where the pre-trained ImageNet model could be much different with the fine-tuned solution, the fine-tuning process is more similar with training from scratch, where large momentum stabilizes the decent directions towards the minimum. An illustration of the difference can be found in Figure 2.

**Connections to early observations on decreasing momentum** Early work (Sutskever et al., 2013) actually pointed out that reducing momentum during the final stage of training allows finer convergence while aggressive momentum would prevent this. They recommended reducing momentum from 0.99 to 0.9 in the last 1000 parameter updates but not disabling it completely. Recent work (Liu et al., 2018; Smith, 2018) showed that a large momentum helps escape from saddle points but can hurt the final convergence within the neighborhood of the optima, implying that momentum should be reduced at the end of training. Liu et al. (2018) find that a larger momentum introduces higher variance of noise and encourages more exploration at the beginning of optimization, and encourages more aggressive exploitation at the end of training. They suggest that at the final stage of the step size annealing, momentum SGD should use a much smaller step size than that of vanilla SGD. When applied to fine-tuning, we can interpret that if the pre-trained model lies in the neighborhood of the optimal solution on the target dataset, the momentum should be small. Our work identifies the empirical evidence of disabling momentum helps final convergence, and fine-tuning on close domains is a good exemplar.

### 3.3 COUPLED HYPERPARAMETERS AND THE VIEW OF EFFECTIVE LEARNING RATE

Now that we had discovered the effect of momentum by fixing other hyperparameters and only allowed momentum to change. But note that the two difficult scenarios shown in Figure 2 (b) and (c) might also be mitigated by increasing or decreasing learning rate. That is, hyperparameters are coupled and varying one hyperparameter can change the optimal values of the other hyperparameters that lead to the best performance. In addition, optimal values of certain hyperparameters depend on the values of other hyperparameters in systematic ways. For example, learning rate is entangled with batch size, momentum and weight decay. There is a notion of *effective learning rate* (ELR) (Hertz et al., 1991; Smith et al., 2018; Smith & Le, 2018) for SGD with momentum: $\eta' = \eta/(1 - m)$, which was shown to be more closely related with training dynamics and final performance rather than $\eta$. The effective learning rate with $m = 0.9$ is 10× higher than the one with $m = 0.0$ if other hyperparameters are fixed, which is probably why we see an increase in optimal learning rate when momentum is disabled in Figure 1(b) and Appendix A.

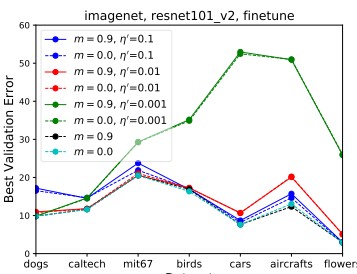

Figure 3: The effect of momentum w/ and w/o fixing ELR $\eta'$. When $\eta'$ is the same, momentum 0 and 0.9 are almost equivalent. If $\eta'$ is allowed to change, there is almost no difference between optimal performance obtained by different $m$.

**It is the effective learning rate that matters for fine-tuning performance** Because hyperparameters are coupled, looking at the performance with only one hyperparameter varied may give a

misleading understanding of the effect of hyperparameters. Therefore, to examine the effect of momentum, we should report the best result obtainable with and without momentum, as long as other hyperparameters explored are sufficiently explored. We re-examine previous experiments that demonstrated the importance of momentum tuning when the ELR $\eta' = \eta/(1-m)$ is held fixed instead of simply fixing learning rate $\eta$. Figure 3 shows that when $\eta'$ is constant, the best performance obtained by $m = 0.9$ and $m = 0$ are almost equivalent when other hyperparameters are allowed to change. However, different ELR does result in different performance, which indicates its importance for the best performance. It explains why the common practice of changing only learning rate generally works, though changing momentum may results in the same result, they both change the ELR. In fact, as long as the initial learning rate is small enough, we can always search for the optimal momentum as it is an amplifier, making the ELR larger by a factor of $1/(1-m)$. Therefore, momentum does determine the search ranges of learning rate.

**Optimal ELR depends on the similarity between source domain and target domain**    Now that we have shown ELR is critical for fine-tuning performance, we are interested in the factors that determine the optimal ELR for a given task. Previous work (Smith & Le, 2018) found that there is an optimum ELR which maximizes the test accuracy. However, the observations are only based on scratch training on small datasets (e.g., CIFAR-10); the relationship between ELR and domain similarity, especially for fine-tuning, is still unexplored. To examine this, we search the best ELR on each fine-tuning task and reports in Fig. 4 the best validation error obtained by each ELR while allowing other hyperparameters to change. It shows the optimal ELR depends on both source domain and target domain. As shown in Fig. 4 (a-c), the optimal ELR for Dogs/Caltech/Indoor are much smaller than these for Aircrafts/Flowers/Cars when fine-tuned from ImageNet pre-trained model. Similar observations can be made on DenseNets and MobileNet. Though the optimal ELR value is different, the relative order of domain similarity is consistent and architecture agnostic. We can also see a smaller ELR works better when source domain and target domain are similar, such as Dogs for ImageNet and Birds for iNat2017 (Fig. 4 (a, d-e)). Interestingly, the optimal ELR for training from scratch is much larger and very similar across different target datasets, which indicates the distance from a random initialization is uniformly similar to different target dataset.

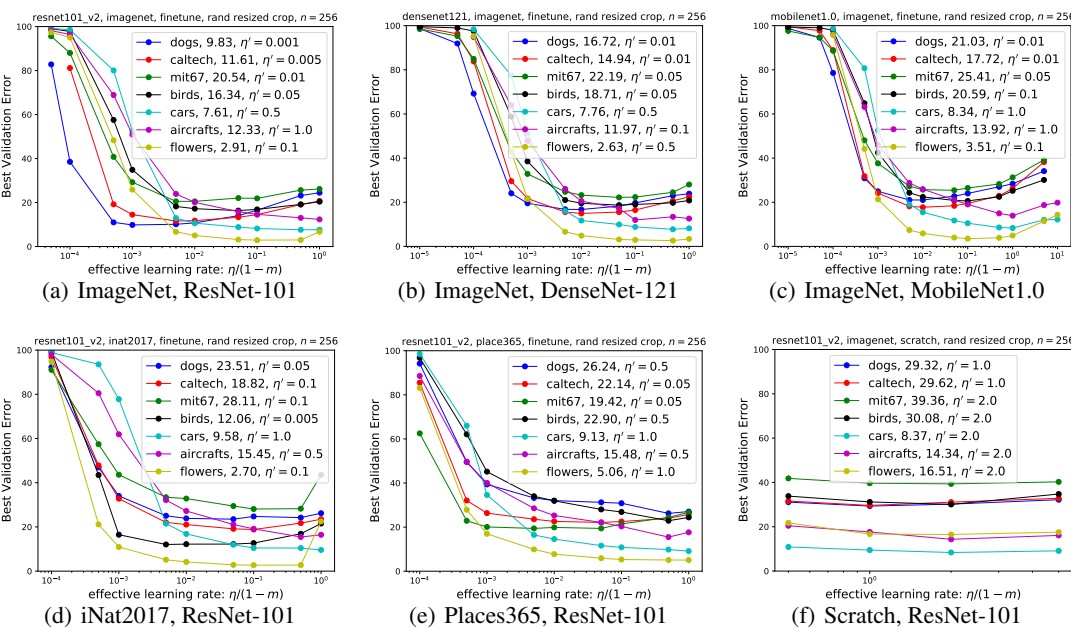

Figure 4: The best validation errors obtained by different ELRs for different source-target domains. Note that the optimal ELR for each target dataset falls in the interior of search space. Each point in (a-e) is the lowest validation error obtained with different weight decay values while ELR is fixed. The first row suggests that the connection between optimal ELR and domain similarity is architecture agnostic. The second row verifies that optimal ELR depends on the similarity between source domain and target domain.

Table 4: The connection between domain similarity and optimal ELR. The values in the second column is provided by Cui et al. (2018), in which JFT pretrained ResNet-101 was used as the feature extractor. Note that neither the pre-trained model or the dataset is released and we cannot calculate the metric for other datasets. In other columns, we calculate domain similarity using ImageNet pre-trained model as the feature extractor. The 1st, 2nd, 3rd and 4th highest scores are color coded. The optimal ELRs are also listed, which corresponds to the values in Fig 4.

| | JFT | ImageNet | | | | | | iNat2017 | | Places365 | |
| | ResNet-101 | ResNet-101 | | DenseNet-121 | | MobileNet | | ResNet-101 | | ResNet-101 | |
| | sim | sim | $\eta'$ | sim | $\eta'$ | sim | $\eta'$ | sim | $\eta'$ | sim | $\eta'$ |
|---|---|---|---|---|---|---|---|---|---|---|---|
| Dogs | 0.619 | 0.862 | 0.001 | 0.851 | 0.01 | 0.852 | 0.01 | 0.854 | 0.05 | 0.856 | 0.5 |
| Caltech | - | 0.892 | 0.005 | 0.881 | 0.01 | 0.878 | 0.01 | 0.871 | 0.1 | 0.888 | 0.05 |
| Indoor | - | 0.856 | 0.01 | 0.850 | 0.05 | 0.839 | 0.01 | 0.843 | 0.1 | 0.901 | 0.05 |
| Birds | 0.563 | 0.860 | 0.05 | 0.842 | 0.05 | 0.849 | 0.1 | 0.901 | 0.005 | 0.861 | 0.5 |
| Cars | 0.560 | 0.845 | 0.5 | 0.831 | 0.5 | 0.830 | 1.0 | 0.847 | 1.0 | 0.864 | 1.0 |
| Aircrafts | 0.556 | 0.840 | 1.0 | 0.817 | 0.1 | 0.831 | 1.0 | 0.846 | 0.5 | 0.853 | 0.5 |
| Flowers | 0.525 | 0.844 | 0.1 | 0.821 | 0.5 | 0.825 | 0.1 | 0.879 | 0.1 | 0.851 | 1.0 |

**Optimal ELR selection based on domain similarity**   Now we have made qualitative observations about the relationship between domain similarity and optimal ELR. A quantitative characterization of the relationship could reduce the hyperparameter search ranges for HPO or even eliminate HPO by accurately predicting hyperparameters. We followed the domain similarity calculation in (Cui et al., 2018) and recalculate similarity scores for all source-target domain pairs. Note the original domain similarity calculation in (Cui et al., 2018) use pre-trained JFT (Sun et al., 2017) models as feature extractor, which are not public available. We alternatively use ImageNet pre-trained model or the source model as feature extractor. As shown in Table 4, there is a good correlation between domain similarity score and the scale of optimal ELR. Generally, the more similar the two domains, the smaller the optimal ELR. Though it is not strictly corresponding to the domain similarity score, the score provides reasonable prediction about the scale of optimal ELR, such as $[0.001, 0.01], [0.01, 0.1], [0.1, 1.0]$ and therefore can reduce the search space for optimal ELR. Based on this correlation, a simple strategy can be developed for optimal ELR selection given a frequently used source model: one can calculate domain similarities and perform exhaustive hyperparameter searches for few reference datasets, including similar and dissimilar datasets. Then given a new dataset to fine-tune, one can calculate the domain similarity and compare with the scores of reference datasets, and choose the range of ELRs with the closest domain similarity.

**Weight Decay and Learning Rate**   The relationship between weight decay and effective learning rate is recently well-studied (van Laarhoven, 2017; Zhang et al., 2018; Loshchilov & Hutter, 2018). It was shown that the effect of weight decay on models with BN layers is equivalent to increasing the ELR by shrinking the weights scales, i.e., $\eta' \sim \eta/\|\theta\|_2^2$. And if the optimal effective learning rate exists, the optimal weight decay value $\lambda$ is inversely related with the optimal learning rate $\eta$. The 'effective' weight decay is $\lambda' = \lambda/\eta$. We show in Figure 5 that the optimal effective weight decay is also correlated with domain similarity.

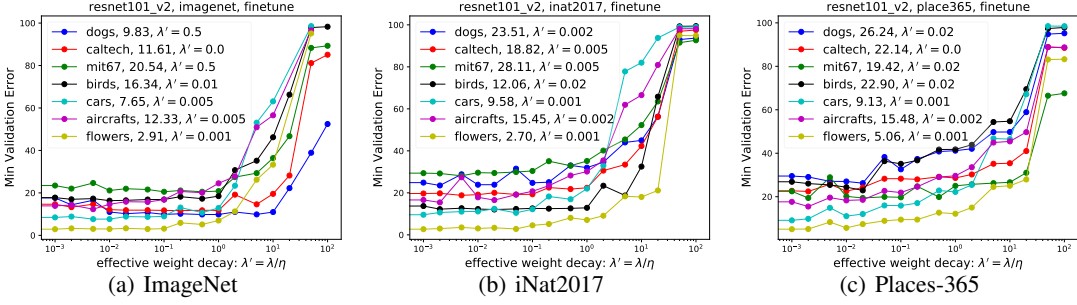

Figure 5: The relationship between optimal effective weight decay and source datasets. The optimal effective weight decay is larger when the source domain is similar with the target domain.

### 3.4 THE CHOICE OF REGULARIZATION

$L_2$ regularization or weight decay is widely used for constraining the model capacity (Hanson & Pratt, 1989; Krogh & Hertz, 1992). Recently Li et al. (2018; 2019) pointed out that standard $L_2$ regularization, which drives the parameters towards the origin, is not adequate in transfer learning. To retain the knowledge learned by the pre-trained model, reference-based regularization was used to regularize the distance between fine-tuned weights and the pre-trained weights, so that the fine-tuned model is not too different from the initial model. Li et al. (2018) propose $L_2$-SP norm, i.e., $\frac{\lambda_1}{2}\|\theta' - \theta_0\|_2^2 + \frac{\lambda_2}{2}\|\theta''\|_2^2$, where $\theta'$ refers to the part of network that shared with the source network, and $\theta''$ refers to the novel part, e.g., the last layer with different number of neurons. While the motivation is intuitive, there are several issues for adopting reference based regularization for fine-tuning:

- Many applications actually adopt fine-tuning on target domains that are quite different from source domain, such as fine-tuning ImageNet models for medical imaging (Mormont et al., 2018; Raghu et al., 2019). The fine-tuned model does not necessarily have to be close with the initial model.

- The scale invariance introduced by Batch Normalization (BN) (Ioffe & Szegedy, 2015) layers enable models with different parameter scales to function the same, i.e., $f(\theta) = f(\alpha\theta)$. Therefore, when $L_2$ regularization drives $\|\theta\|_2^2$ towards zeros, it could still have the same functionality as the initial model. On the contrary, a model could still be different even when the $L_2$-SP norm is small.

- $L_2$-SP regularization would constrain $\theta''$ to be close to $\theta_0$, so that $\|\theta\|_2^2$ is relatively stable in comparison with $L_2$ regularization. Given that ELR is approximately proportional to $\eta/\|\theta\|_2^2$ and a smaller ELR is beneficial for fine-tuning from similar domains, it may explain why $L_2$-SP provides better performance. If this is true, then by decreasing the initial ELR, $L_2$-norm may function the same.

To examine these conjectures, we revisited the work of (Li et al., 2018) with additional experiments. To show the effectiveness of $L_2$-SP norm, the authors conducted experiments on datasets such as Dogs, Caltech and Indoor, which are all close to the source domain (ImageNet or Places-365). We extend their experiments by fine-tuning on both "similar" and "dissimilar" datasets, including Birds, Cars, Aircrafts and Flowers, with both $L_2$ and $L_2$-SP regularization (details in Appendix D). For fair comparison, we perform the same hyperparameter search for both methods. As expected, Table 5 shows that $L_2$ regularization is very competitive with $L_2$-SP on Birds, Cars, Aircrafts and Flowers, which indicates that reference based regularization may not generalize well for fine-tuning on dissimilar domains.

Table 5: The average class error of (Li et al., 2018) and the extension of their experiments of on "dissimilar" datasets. The *italic* datasets and numbers are our experimental results. Note that the original Indoor result is fine-tuned from Places-365, while we fine-tune just from ImageNet pre-trained models.

| Method | Dogs | Caltech | Indoor | *Birds* | *Cars* | *Flowers* | *Aircrafts* |
|---|---|---|---|---|---|---|---|
| $L_2$ (Li et al., 2018) | 18.6 | 14.7 | 20.4 | - | - | - | - |
| $L_2$-SP (Li et al., 2018) | **14.9** | **13.6** | **15.8** | - | - | - | - |
| $L_2$ with HPO | *16.79* | *14.98* | *23.00* | *22.51* | *10.10* | *5.70* | ***13.03*** |
| $L_2$-SP with HPO | ***13.86*** | ***14.45*** | ***21.77*** | ***22.32*** | ***9.59*** | ***5.28*** | *13.31* |

We also check the change of regularization terms during training for both methods as well as their best hyperparameters. As shown in Figure 6, the $L_2$ regularization usually decrease the weights norm more aggressively, depending on the value of $\lambda$, while $L_2$-SP regularization keeps the norm less changed. We can see that the optimal learning rate of $L_2$ regularization is mostly smaller than $L_2$-SP, which may compensate for the decreased weight norm or increased ELR. Interestingly, for Dogs dataset, both regularization terms grow much larger after a few iterations and then become stable, which means constraining the weights to be close to initialization is not necessarily the reason for $L_2$-SP to work even for close domains. It also seems contradicting to previous finding (Zhang et al., 2018) that weight decay functions as increasing ELR by decreasing weight norms. However, it

might be reasonable as large norm actually decreases the ELR, which could be helpful due to the close domain similarity between Dogs and ImageNet.

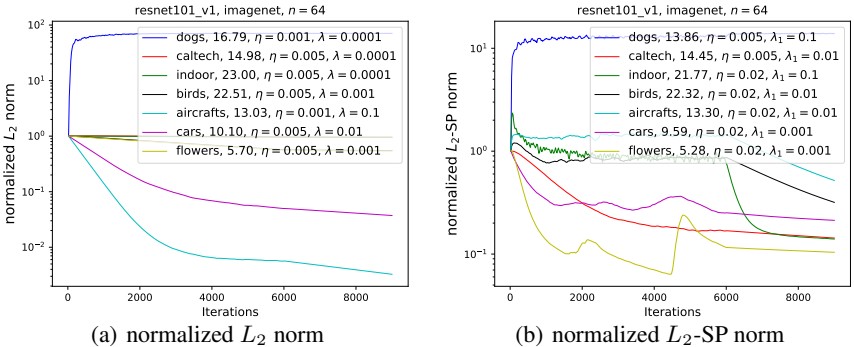

(a) normalized $L_2$ norm          (b) normalized $L_2$-SP norm

Figure 6: The normalized $L_2$ norm and $L_2$-SP norm during training. The $y$-axis is the relative change of the regularization term in comparison to the initial value, i.e., $\|\theta_t\|_2^2/\|\theta_0\|_2^2$ for $L_2$ norm and $(\lambda_1\|\theta_t' - \theta_0\|_2^2 + \lambda_2\|\theta_t''\|_2^2)/(\lambda_2\|\theta_0''\|_2^2)$ for $L_2$-SP norm. Optimal hyperparameters are also given in the legend. Note that experiment uses batch size 64 instead of 256, which results in smaller optimal learning rate comparing to previous result.

## 4 DISCUSSION

The two extreme ways for selecting hyperparameters—performing exhaustive hyperparameter search or taking ad-hoc hyperparameters from scratch training—could be either too computationally expensive or yield inferior performance. Different from training from scratch, where the default hyperparameter setting may work well for random initialization, the choice of hyperparameters for fine-tuning is not only dataset dependent but is also influenced by the similarity between the target and source domains. The rarely tuned momentum value could also improve or impede the performance when the target domain and source domain are close given insufficiently searched learning rate. These observations connect with previous theoretical works on decreasing momentum at the end of training and effective learning rate. We further identify that the optimal effective learning rate correlates with the similarity between the source and target domains. With this understanding, one can significantly reduce the hyperparameter search space. We hope these findings could be one step towards better and efficient hyperparameter selection for fine-tuning.

### ACKNOWLEDGMENTS

The authors would like to thank all anonymous reviewers for their valuable feedback.

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

## A  THE EFFECTIVENESS OF MOMENTUM

**Searching for Optimal Momentum**    To check the effectiveness of momentum on fine-tuning, we can search the best momentum values for fine-tuning with fixed learning rate but different weight decay and batch size. Taking Birds dataset as an example, Figure 7 provides the convergence curves for the results shown in Figure 1(a), which shows the learning curves of fine-tuning with 6 different batch sizes and weight decay combinations. Zero momentum outperforms the nonzero momentum in 5 out of 6 configurations.

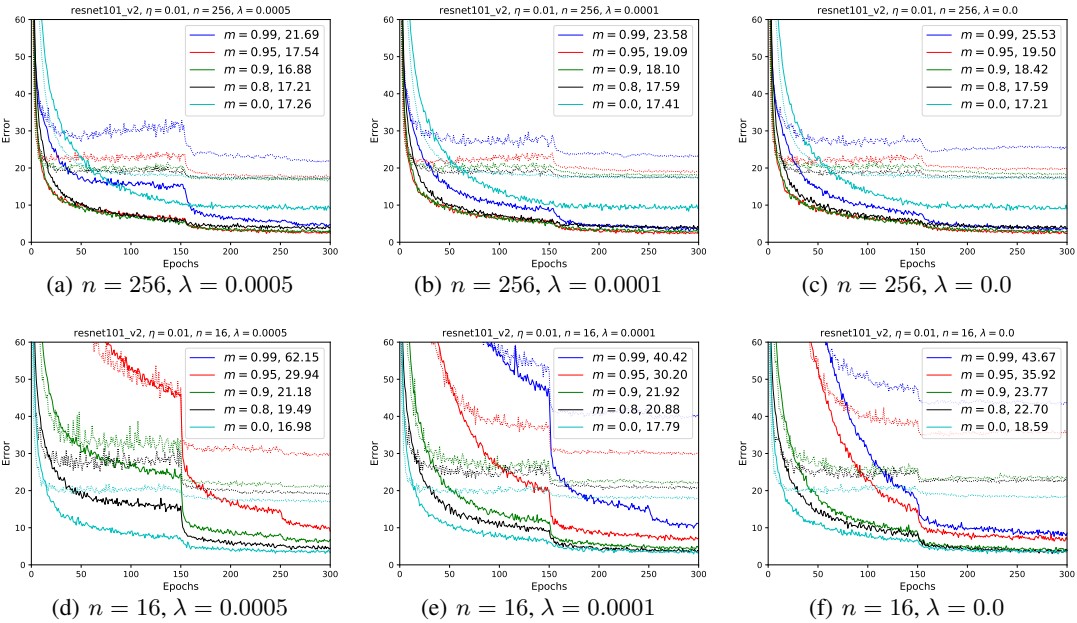

Figure 7: Searching for the optimal momentum on Birds dataset with fixed learning rate and weight decays. The solid lines are training errors and the dashed lines are validation errors.

**Effective learning rate increases after disabling momentum.**    Figure 8 compares the performance of with and without momentum for Dogs dataset with a range of different learning rates. Note that the learning rate with similar performance generally increases 10x after changing $m$ from 0.9 to 0.0, which is coherent with the rule of effective learning rate $\eta' = \eta/(1 - m)$. Same observations can be made on other datasets as shown in Figure 9.

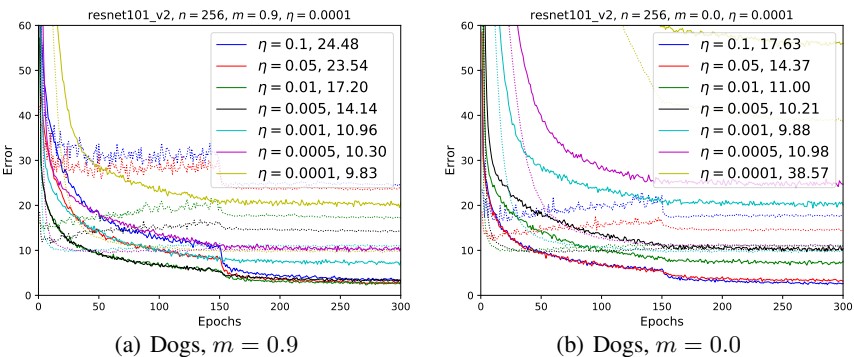

Figure 8:  The effect of momentum when learning rate is allowed to change. The learning rate for the best performance increases 10x after changing $m$ from 0.9 to 0.0, which is coherent with the rule of effective learning rate. Note that weight decay $\lambda$ is fixed at 0.0001.

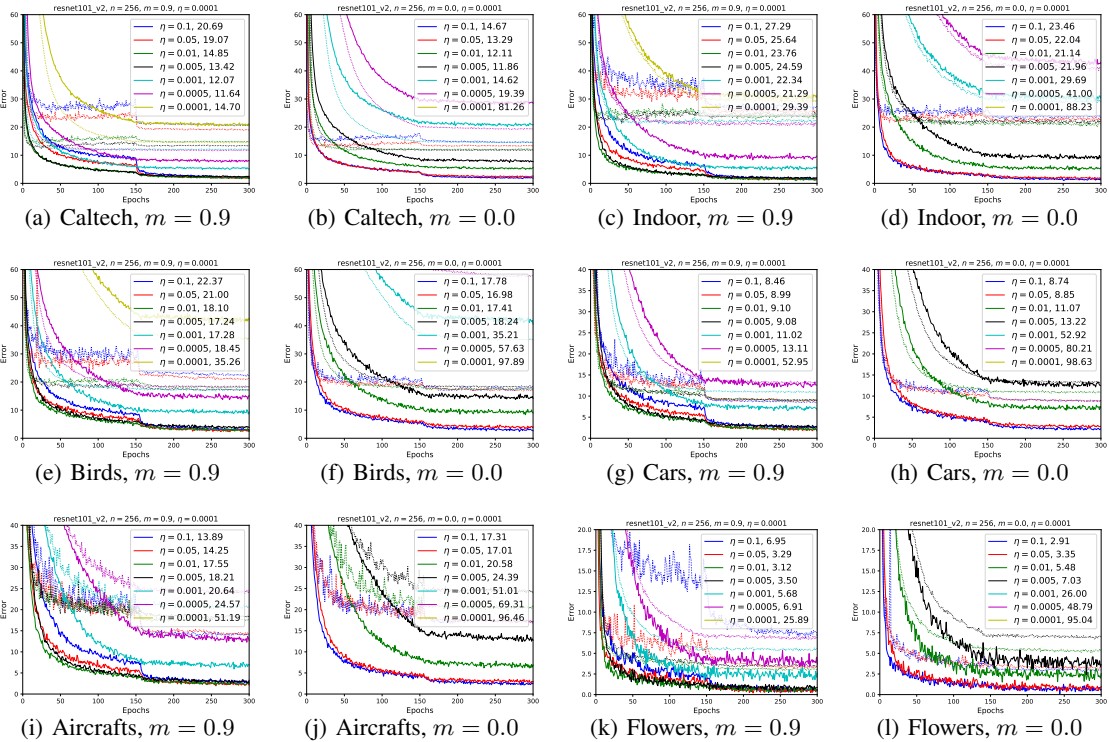

Figure 9: The effect of momentum when learning rate is allowed to change (Figure 8 continued). The learning rate for the best performance increases 10x after changing $m$ from 0.9 to 0.0, which is coherent with the rule of effective learning rate.

## B  DOMAIN SIMILARITY

The domain similarity calculation based on Earth Mover Distance (EMD) is introduced in the section 4.1 of (Cui et al., 2018)[4]. Here we briefly introduce the steps. In (Cui et al., 2018), the authors first train ResNet-101 on the large scale JFT dataset (Sun et al., 2017) and use it as a feature extractor. They extracted features from the penultimate layer of the model for each image of the training set of the source domain and target domain. For ResNet-101, the length of the feature vector is 2048. The features of images belonging to the same category are averaged and $g(s_i)$ denotes the average feature vector of $i$th label in source domain $S$, similarly, $g(t_j)$ denotes the average feature vector of $j$th label in target domain $T$. The distance between the averaged features of two labels is $d_{i,j} = \|g(s_i) - g(t_j)\|$. Each label is associated with a weight $w \in [0, 1]$ corresponding to the percentage of images with this label in the dataset. So the source domain $S$ with $m$ labels and the target domain $T$ with $n$ labels can be represented as $S = \{(s_i, w_{s_i})\}_{i=1}^m$ and $T = \{(t_j, w_{t_j})\}_{i=1}^n$. The EMD between the two domains is defined as

$$d(S, T) = \text{EMD}(S, T) = \frac{\sum_{i=1,j=1}^{m,n} f_{i,j} d_{i,j}}{\sum_{i=1,j=1}^{m,n} f_{i,j}} \tag{3}$$

where the optimal flow $f_{i,j}$ corresponds to the least amount of total work by solving the EMD optimization problem. The domain similarity is defined as

$$\text{sim}(S, T) = e^{-\gamma d(S,T)} \tag{4}$$

where $\gamma$ is 0.01. Note that the domain similarity value is not ranging from 0 to 1.

Due to the unavailability of the large-scale JFT dataset (300x larger than ImageNet) and its pre-trained ResNet-101 model, we cannot use it for extracting features for new datasets, such as Caltech256 and

---

[4]The extracted features and code are available in https://github.com/richardaecn/cvpr18-inaturalist-transfer

MIT67-Indoor. Instead of using the powerful feature representation, we use our pre-trained ImageNet model (ResNet-101) as the feature extractor. Table 4 compares the domain similarities calculated by different pre-trained models and we can see some consistent patterns across different architectures: e.g., The 1st and 2nd highest similarity scores are Caltech and Dogs regardless of architectures; the 3rd and 4th highest similarity scores refers to Birds and Indoor; the most dissimilar datasets are Cars, Aircrafts and Flowers, though the relative orders for them are not exactly the same. Besides using fixed feature extractor, an alternative way is to use the source domain model directly as the feature extractor for both source domain and target domain, which may captures the transfer learning process more precisely than a uniform feature extractor.

## C   THE EFFECTIVENESS OF BN MOMENTUM

Kornblith et al. (2019) conducted extensive fine-tuning experiments with different hyperparameters. One observation they made is that the momentum parameter of BN layer is essential for fine-tuning. They found it useful to decrease the BN momentum parameter from its ImageNet value to $\max(1 - 10/s, 0.9)$ where $s$ is the number of steps per epoch. This will change the default BN momentum value (0.9) when $s$ is larger than 100, but it only applies when the dataset size is larger than 25.6K with batch size 256. The maximum data size used in our experiments is Caltech-256, which is 15K, so this strategy seems not applicable.

We further validate the effect of BN momentum by performing a similar study as to ELR. The goal is to identify whether there is an optimal BN momentum for a given task. For each dataset, we fine-tune the pre-trained model using previously obtained best hyperparameters and only vary BN momentum. In addition to the default value 0.9, we also set it to 0.0, 0.95 and 0.99. The rational is that if BN momentum is a critical hyperparameter, we should expect significant performance differences when the value is changed from the optimal value. As shown in Figure 10, we can see $m_{bn} = 0.99$ slightly improves the performance for some datasets, however, there is no significant performance difference among values greater than 0.9. One hypothesis is that similar domains will share similar BN parameters and statistics, BN momentum may affect the parameter adaptation. More investigation is still needed to fully understand its effectiveness.

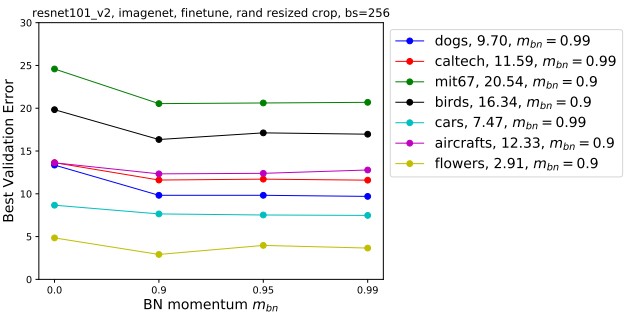

Figure 10: Performance of different BN momentum for each dataset with existing optimal hyperparameters.

## D   EXPERIMENTAL SETTINGS FOR COMPARISON OF $L_2$ AND $L_2$-SP

The experiments in Section 3.4 is based the code[5] provided by (Li et al., 2018). The base network is ImageNet pretrained ResNet-101-V1. The model is fine-tuned with batch size 64 for 9000 iterations, and learning rate is decayed once at iteration 6000. Following the original setting, we use momentum 0.9. We performed grid search on learning rate and weight decay, with the range of $\eta : \{0.02, 0.01, 0.005, 0.001, 0.0001\}$ and $\lambda_1 : \{0.1, 0.01, 0.001, 0.0001\}$, and report the best average class error (1 - average accuracy) for both methods. For $L_2$-SP norm, we follow the authors' setting to use constant $\lambda_2 = 0.01$. Different with the original setting for $L_2$ regularization, we set $\lambda_2 = \lambda_1$ to simulate normal $L_2$-norm.

---

[5] https://github.com/holyseven/TransferLearningClassification

# E    DATA AUGMENTATION

Data augmentation is an important way of increasing data quantity and diversity to make models more robust. It is even critical for transfer learning with few instances. The effect of data augmentation can be viewed as a regularization and the choice of data augmentation can be also viewed as a hyperparameter. Most current widely used data augmentation methods have verified their effectiveness on training ImageNet models, such as random mirror flipping, random rescaled cropping[6], color jittering and etc (Szegedy et al., 2015; Xie et al., 2018).

Do these methods transfer for fine-tuning on other datasets? Here we compare three settings for data augmentation with different momentum settings: 1) random resized cropping: our *default* data augmentation; 2) random cropping: the same as *standard* data augmentation except that we use random cropping with fixed size; 3) random flip: simply random horizontal flipping. The training and validation errors of fine-tuning with different data augmentation strategies and hyperparameters are shown in Figure 11 and Figure 12.

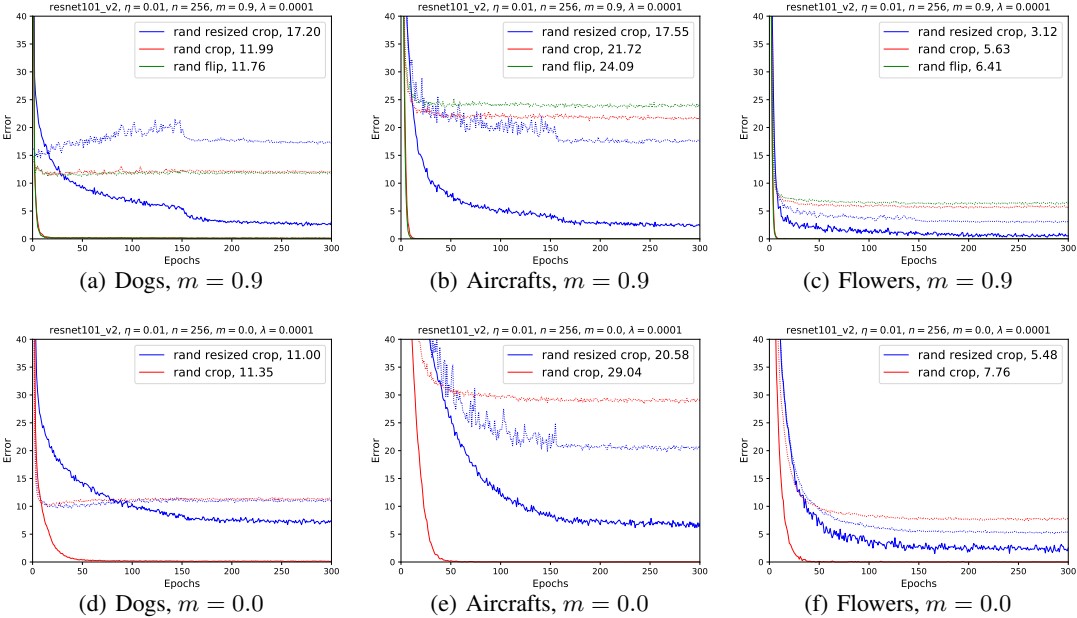

Figure 11: Fine-tuning with different data augmentation methods and hyperparameters. Dashed curves are the validation errors. Strong data augmentation is harder to train as it converge slowly and needs more number of epochs to observe the advanced performance on datasets such as Aircrafts. Simple data augmentation (red curves) converges much faster in training error. Strong data augmentation (blue curves) overfits the Dogs dataset with default hyperparameter but performs well with $m = 0$.

**The effect of data augmentation is dataset dependent and is also influenced by other hyperparameters**    The first row in Figure 11 shows that advanced data augmentation with default hyperparameters ($m = 0.9$ and $\eta = 0.01$) leads to overfitting for Dogs while generalize better on Aircrafts and Flowers. Similar observations can be made in Figure 12. However, when momentum is disabled, the overfitting disappears for Dogs and Caltech. This is explainable since random resized cropping adds more variance to the gradient direction, and disabling momentum will lead to a smaller ELR which will be helpful for fine-tuning from a similar domain. On the other hand, the performance of random cropping decreases when momentum is disabled. As random cropping produces training samples with less variation than random resized cropping, disabling momentum or decreasing the ELR might lead to underfitting or stucking in poor local minima. This can be mitigated as we increase the learning rate for random cropping, which adds variation to the gradients. As shown in Table 6,

---

[6]Randomly crop a rectangular region with aspect ratio randomly sampled in [3/4, 4/3] and area randomly sampled in [8%, 100%] (Szegedy et al., 2015)

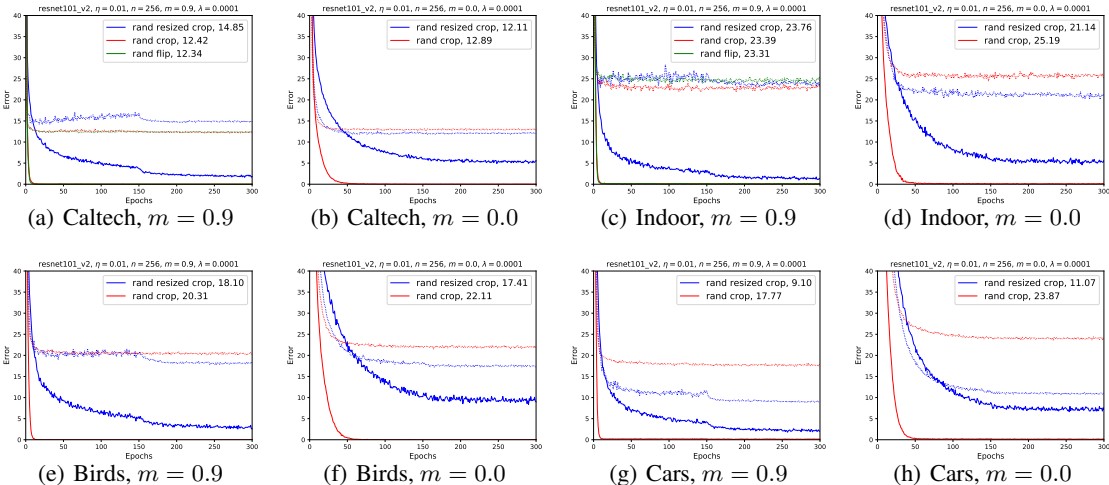

Figure 12: Comparison of data augmentation methods with different momentum values (Figure 11 continued). The other hyperparameters are: $n = 256$, $\eta = 0.01$ and $\lambda = 0.0001$.

when learning rate increased fro 0.01 to 0.05, disabling momentum shows better performance than nonzero momentum on datasets that are close, similar to previous findings with random resized cropping.

Table 6: Comparison of data augmentation methods with different momentum values. The rest of the hyperparameters are: $n = 256$ and $\lambda = 0.0001$.

| Data Augmentation | $m$ | $\eta$ | Dogs | Caltech | Indoor | Birds | Cars | Flowers | Aircrafts |
|---|---|---|---|---|---|---|---|---|---|
| Rand resized crop | 0.9 | 0.01 | 17.20 | 14.85 | 23.76 | 18.10 | **9.10** | **3.12** | **17.55** |
| | 0 | 0.01 | **11.00** | **12.11** | **21.14** | **17.41** | 11.06 | 5.48 | 20.58 |
| Rand crop | 0.9 | 0.01 | 11.99 | **12.42** | **23.39** | 20.31 | 17.77 | 5.63 | 21.72 |
| | 0 | 0.01 | **11.35** | 12.89 | 25.19 | 22.11 | 23.87 | 7.76 | 29.04 |
| | 0.9 | 0.05 | 16.85 | 14.80 | 23.46 | **18.81** | 13.70 | **4.85** | **17.64** |
| | 0 | 0.05 | **11.79** | **12.52** | **23.24** | 20.69 | 20.00 | 7.06 | 23.43 |

## F    SOURCE DOMAINS

**Pre-trained models**    For most of our experiments, we use the pre-trained ResNet-101_v2 model from the model zoo of MXNet GluonCV [7]. To get the pre-trained models for iNat-2017 and Places-365, we fine-tune from the ImageNet pre-trained model with the *default* fine-tuning hyperparameters for 60 epochs, where learning rate is decayed at epoch 45 by a factor of 10. Table 7 illustrates the Top-1 errors of each pre-trained model on their validation sets.

Table 7: The Top-1 error of ResNet-101 pre-trained on different source dataset.

| Dataset | class | Top-1 error |
|---|---|---|
| ImageNet | 1000 | 21.4 |
| iNat2017 | 5,089 | 32.2 |
| Places-365 | 365 | 31.5 |

---

[7] https://gluon-cv.mxnet.io/model_zoo/classification.html

**Training from Scratch with HPO** The default hyperparameters for training from scratch are $\eta = 0.1$, $\lambda = 0.0001$, $m = 0.9$ and $n = 256$. We train 600 epochs, and decay the learning rate at epoch 400 and 550 by a factor of 10. To perform Hyperparameter Optimization (HPO), we search hyperparameters in the following space: $\eta \in [0.1, 0.2, 0.5]$ and $\lambda \in [0.0001, 0.0005]$. Figure 13 shows the training/validation errors of training from scratch on each dataset with different learning rate and weight decay. We observe weight decay 0.0005 consistently performs better than 0.0001.

**Insufficient hyperparameter search may lead to miss-leading conclusion** To show the importance of hyperparameter tuning, Table 8 compares the performance with and without hyperparameter tuning for both fine-tuning and training from scratch tasks. With the default hyperparameters, some inappropriate conclusions might be made, e.g., "there is significant gap between fine-tuning and training from scratch", "fine-tuning always surpass training from scratch" or "fine-tuning from iNat cannot beat the performance of ImageNet". However, with HPO, those statements may not be valid. For example, training from scratch surpass the default fine-tuning result on Cars and Aircrafts and the gap between fine-tuning and training from scratch is much smaller. Previous studies (Kornblith et al., 2019; Cui et al., 2018) also identified that datasets like Cars and Aircrafts do not benefit too much from fine-tuning.

Table 8: Comparison of default hyperparameters and HPO for both fine-tuning (FT) and training from scratch (ST) tasks. FT Default and ST Default use their *default* hyperparameters, respectively. HPO refers to the finding the best hyperparameters with grid search.

| Method | Source | Dogs | Caltech | Indoor | Birds | Cars | Aircrafts | Flowers |
|--------|--------|------|---------|--------|-------|------|-----------|---------|
| FT Default | ImageNet | **17.20** | **13.42** | 23.76 | 18.10 | **9.10** | **17.55** | **3.12** |
| FT Default | iNat2017 | 24.74 | 20.12 | 30.73 | **14.69** | 11.16 | 19.86 | 3.19 |
| FT Default | Places-365 | 30.84 | 22.53 | **22.19** | 27.72 | 11.06 | 21.27 | 5.66 |
| ST Default | - | 38.26 | 36.21 | 45.28 | 43.72 | 16.73 | 26.49 | 22.88 |
| FT HPO | ImageNet | **9.83** | **11.61** | 20.54 | 16.34 | **7.61** | **12.33** | 2.91 |
| FT HPO | iNat2017 | 23.51 | 18.82 | 28.11 | **12.06** | 9.58 | 15.45 | **2.70** |
| FT HPO | Places-365 | 26.24 | 22.14 | **19.42** | 22.90 | 9.13 | 15.48 | 5.06 |
| ST HPO | - | 29.32 | 29.62 | 39.36 | 30.08 | 8.37 | 14.34 | 16.51 |

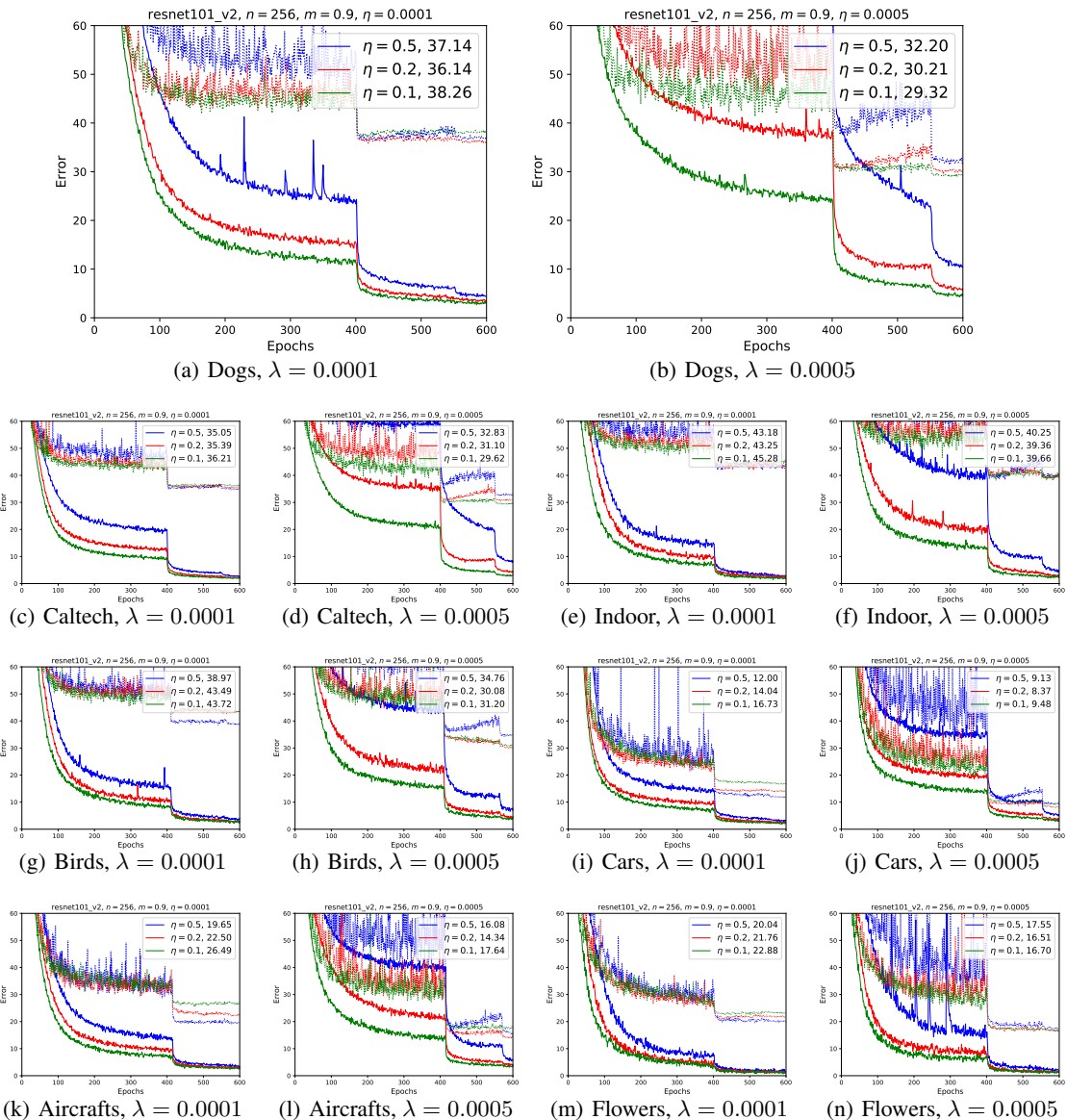

Figure 13: Training from scratch with various learning rate and weight decay. The batch size is 256 and the momentum is 0.9. The solid curves are training error and the dashed lines are valdiation error.

