# OpenReview forum: "Rethinking the Hyperparameters for Fine-tuning"
_ICLR.cc/2020/Conference — Accept (Poster)_

### Official Review · AnonReviewer2 · 2019-10-19
**Official Blind Review #2**

**Rating:** 6

**Review:**

This paper studies the role of different hyperparameters in finetuning image recognition models on new target tasks. The authors run a large set of experiments and show that, perhaps non-surprisingly, hyperparameters matter. In particular, they show that momentum, which is typically ignored in finetuning, is quite important, and that the momentum values  that work well depend on the similarity between the source and target datasets. They also show important correlations between momentum, learning rate, and weight decay.
Overall, despite some issues detailed below, the paper is clearly written, presents a coherent story, and its conclusions will be useful to the community.

Comments:

1. My main concern about this paper relates to the importance of momentum. The authors argue that this hyperparameter is "critical for fine-tuning performance". However, they later show that in fact what matters is the ratio between the learning rate (LR) and the momentum. In this case, it might be justified to fix the momentum value and only modify the LR, as often done.

2. The EMD values of Birds, Cars and Aircrafts are within 0.7 points of each other (while Dogs is much higher and Flowers is quite lower). Although I am not too familiar with this method, I find it somewhat hard to believe that these small differences explain the error differences on Table 2.

3. While the paper is fairly clear in writing, the figures (e.g., fig. 3 and 4) are extremely hard to read on print, and thus hard to draw conclusions from. Figure 4 is confusing also on screen.

4. To promote reproducibility, it would be better to report in this kind of research validation rather than test results. There is some confusion in Figure 4, the axes say validation error, while the caption says test error, but in the other figures test results are reported.

Minor:

1. The authors say in the intro "Even when there is enough training data, fine-tuning is still preferred as it often reduces training time significantly (He et al., 2019).", but later make a somewhat contradictory claim: "He et al. (2019) questioned whether ImageNet pre-training is necessary for training object detectors. They find the solution of training from scratch is no worse than the fine-tuning counterpart as long as the target dataset is large enough.".

2. A couple of typos around the paper:
- section 2: "However, most of these advances on hyperparameter tuning are designed *from* training from scratch" (should be "for")
- The first sentence of 3.3 is ungrammatical



**Experience Assessment:**

I have published one or two papers in this area.

**Review Assessment: Checking Correctness Of Derivations And Theory:**

I assessed the sensibility of the derivations and theory.

**Review Assessment: Checking Correctness Of Experiments:**

I assessed the sensibility of the experiments.

**Review Assessment: Thoroughness In Paper Reading:**

I read the paper thoroughly.

---

> ### Author Response · Authors · 2019-11-15
> **our response**
>
> Thanks for your positive review! In our revised version, we have fixed the typos and made the figures easier to read and adjusted our narratives. Below are our response to your questions:
>
> Q1: “My main concern about this paper relates to the importance of momentum”
>
> A:  We agree that simply emphasizing the importance of momentum without context may be miss leading. As we agree that learning rate is a “critical” hyperparameter, the aim of the first part is to show momentum is also an equivalent “critical” hyperparameter, rather than common belief that the default 0.9 is the best value or there is no effect when momentum is tuned. We would like to correct the belief that only LR is “critical” for fine-tuning.
>
> In fact, as long as the initial learning rate is fixed and small enough, we can always search for the optimal momentum instead. Momentum acts as a learning rate amplifier, making the ELR larger by a factor of 1/(1-m). This shares the same spirit as [Smith et al, 2018] where increasing batch size during training has the same effect as decreasing learning rate. When momentum is fixed, it also determine the search ranges of learning rate. Tiny changes of momentum from 0.99 to 0.9 will have significant effect on the learning rate search ranges.
>
> Q2: “The EMD values of Birds, Cars and Aircrafts are within 0.7 points of each other...I find it somewhat hard to believe that these small differences explain the error differences on Table 2”
>
> A: Though difference of the similarity scores are not dramatic, the relative order is more meaningful. The similarity based on EMD might not be the optimal measurement, but the difference of optimal hyperparameters for those datasets are clearly shown in Figure 4. In Appendix C of the revised version, we provide the detailed steps for calculating the EMD based domain similarity and provide our scores for all datasets. The raw EMD for Birds and Cars are 15.08 and 16.82 and their similarity scores after processing are 0.8600 and 0.8452. As shown in Figure 4(a), the optimal ELR for Birds and Cars are 0.05 and 0.5, respectively. With the same learning rate 0.01, both momentum 0 and 0.9 has similar performance for Birds as the ELR are all close to the optimal one. While for Cars dataset, momentum 0.9 makes 0.1 ELR, which is closest to the optimal ELR 0.5.
>
> Q3: “it would be better to report in this kind of research validation rather than test results”
>
> A: We agree. We use the test data of these datasets for validation and report the validation error. We have corrected the axes in Figure 4 and make them more consistent. As we noted in section 3.1, our goal is not getting STOA performance on these datasets. For each trial of hyperparameter configuration, we report the performance of the last epoch after training rather than selecting the best one during training. The curves are only used for monitoring and comparison.
>
> [1] Smith et al, Don’t decay the learning rate, increase the batch size, ICLR 2018

---

### Official Review · AnonReviewer1 · 2019-10-21
**Official Blind Review #1**

**Rating:** 6

**Review:**

This paper provides extensive experimental results to investigate the influence of hyper-parameters on fine-tuning and challenges several commonly-held beliefs. The hyper-parameters of training from scratch does not always perform well when applied to fine-tuning. Furthermore, current L_2-SP regularization is not necessarily helpful when the domain discrepancy is large.
The authors discover that the optimal momentum value is closely related to domain similarity. For similar target datasets, 0 momentum is a better choice than 0.9, since it potentially allows better convergence. Similar to training from scratch, the actual effect at play is the effective learning rate and ‘effective’weight decay. This further involves the coupling of hyper-parameters.
Different from the commonly-held belief, the L_2-SP regularization does not always perform better than L_2. When domain discrepancy is large, the regularization effect will be worsened.
This paper is well-written and makes several interesting discoveries. My question for the authors is as follows:
In the momentum section, the authors postulate that for more similar target datasets, smaller momentum performs better. Here, the similarity is quantified by EM distance defined in the feature space. However, for the five datasets provided, the similarity of them are really close, making this claim less convincing. The conclusion is reasonable, but the authors may need a more reliable method to compare the similarity between datasets.


**Experience Assessment:**

I have published one or two papers in this area.

**Review Assessment: Checking Correctness Of Derivations And Theory:**

I assessed the sensibility of the derivations and theory.

**Review Assessment: Checking Correctness Of Experiments:**

I carefully checked the experiments.

**Review Assessment: Thoroughness In Paper Reading:**

I read the paper thoroughly.

---

> ### Author Response · Authors · 2019-11-15
> **our response**
>
> Thanks for your positive review! We have updated our paper and added Appendix C regarding the concerns on similarity calculation.
>
> Q1: “for the five datasets provided, the similarity of them are really close, making this claim less convincing...the authors may need a more reliable method to compare the similarity between datasets”
>
> A: The original Earth Mover’s Distance is much larger, e.g., the raw EMD value for Birds and Cars are 15.08 and 16.82. We follow the method of Cui et al, 2018 by using the scaled exponential value as similarity, and their similarity scores after processing are 0.8600 and 0.8452. We introduce this process in Appendix C of the revised version and made further comparison with the original approach.
> Different with Cui et al, 2018 that calculated the similarity with ResNet101 pre-trained on JFT, which is not public available. We use the source model as the feature extractor. We find the scale of similarity is a bit different with the original paper. But the relative similarities to ImageNet is almost the same, such as Dogs is still more similar to ImageNet than Birds or Cars and it is consistent across different architectures.
> We find that the similarities correlates with the scale of optimal ELR pretty well, though it is not a strict correspondence. This is already useful for reducing the hyperparameter search range in a dataset dependent way. We provide a heuristic approach for exploiting this correlation. We believe more reliable and accurate similarity calculation method will be very useful for efficient optimal hyperparameters prediction for fine-tuning and our findings are one step towards it.

---

### Official Review · AnonReviewer3 · 2019-10-24
**Official Blind Review #3**

**Rating:** 6

**Review:**

This submission studies the problem of transfer learning and fine tuning. This submission proposes four insights: Momentum hyperparameters are essential for fine-tuning; When the hyperparameters satisfy some certain relationships, the results of fine-tuning are optimal; The similarity between source and target datasets influences the optimal choice of the hyperparameters; Existing regularization methods for DNN is not effective when the datasets are dissimilar. This submission provides multiple experiments to support their opinion.

Pros:
+  This submission provides interesting facts that are omitted in previous research works.
+  This submission examines the previous theoretical results in empirical setting and finds some optimal hyperparameter selection strategies.
+  This submission provides many experiment results of fine-tuning along with its choice of hyperparameters that could be taken as baselines in future researches.

Cons:
-	All experiments results are based on same backbone, which makes all discoveries much less reliable. More experiments on other backbones are necessary. Furthermore, this submission claims that the regularization methods such as L2-SP may not work on networks with Batch Normalization module. But there is no comparison on networks without BN.
-	Providing a complete hyperparameter selecting strategy for fine-tuning could be an important contribution of this submission. I suggest authors to think about it.
-	This submission claim that the choice of hyperparameters should depend on similarity of different domains. But this submission does not propose a proper method for measure the similarity or provide detailed experiments on previous measurements.
-	It seems that the MITIndoors Dataset is not similar with ImageNet from the semantic view. This submission does not provide similarity measurement between these datasets. Why the optimal momentum is 0?
-	The effective learning rate and ‘effective’ weight decay are not first given in this submission. This makes the novelty of this submission relatively weak. Authors only test these strategies in fine-tuning setting and find that they also work with a different initialization.
-	It seems that merely searching for learning rate and weight decay hyperparameters (as Kornblith et al. (2018) did) on a fixed momentum is Ok if there is a most effective relationship between learning rate and momentum. So the discoveries in the first part that a 0 momentum can be better is based on a careless search of learning rates?
-	This submission omits that Kornblith et al. (2018) also referred to the fact that the momentum parameter of BN is essential for fine-tuning and provided a strategy in section A.5. Discussion about this strategy will make this submission more complete.

This submission gives important discoveries about the hyperparameter choice in the fine-tuning setting. But there are several flaws in this submission. I vote for rejecting this submission now but I expect authors to improve the submission in the future version.


**Experience Assessment:**

I have published one or two papers in this area.

**Review Assessment: Checking Correctness Of Derivations And Theory:**

I carefully checked the derivations and theory.

**Review Assessment: Checking Correctness Of Experiments:**

I carefully checked the experiments.

**Review Assessment: Thoroughness In Paper Reading:**

I read the paper thoroughly.

---

> ### Author Response · Authors · 2019-11-15
> **our response 2/2**
>
> Q6: “The effective learning rate and ‘effective’ weight decay are not first given in this submission. This makes the novelty of this submission relatively weak.”
>
> A: Yes, we are not claiming we discovered effective learning rate/weight decay, which states the relationship of different hyperparameters. As opposed to previous work which mostly studied ELR based on experiments of training small models from scratch, we identify the optimal ELR varies a lot due to different initialization in fine-tuning. The novelty is to identify how domain similarity affect the optimal hyperparameter selection, not only about changing learning rate. To the best of our knowledge, this is the first to show the optimal ELR is initialization dependent with practical selection strategy. We also identify caveats with accepted wisdoms such as fixing the momentum to 0.9 for fine-tuning.
>
> Q7: “It seems that merely searching for learning rate and weight decay hyperparameters (as Kornblith et al. (2018) did) on a fixed momentum is Ok if there is a most effective relationship between learning rate and momentum.“
>
> A: As we stated in section 3.3 “it is the effective learning rate that matters for the best performance. It explains why the common practice of changing only learning rate generally works, though changing momentum may result in the same effect”. We acknowledge the effectiveness of the common practice of searching only learning rate and weight decay, but providing better understanding that fixing momentum is not because of 0.9 is the best value or only learning rate matters. Note that It is difficult to find a range for hyper-parameter search of learning rates for fine-tuning. This is trickier than the case of training from scratch because learning rate for fine-tuning also depends on the distance between the source and target domain.
>
> The discovery in the first part is based on a small local search of the default learning rate (0.01 and 0.005). The aim of the first part is to show that simply tuning momentum with default learning rate can also change the performance dramatically, as opposed to common belief that momentum is not necessary to tune as it is not “critical” or 0.9 is the best value. In fact, as long as the initial learning rate is small enough, we can also find the optimal ELR by tuning momentum, as momentum is an amplifier that makes the ELR larger by a factor of 1/(1-m). Therefore, momentum determines the search ranges of learning rate.
> Similar to [Smith et al, 2018], which shows that increasing batch size has the same effect of decaying learning rate. Though it turns out that common practice that simply tuning momentum works, we provided an alternative view that one can also simply tune momentum as lone as the learning rate is small enough.
>
> Q8: “This submission omits that Kornblith et al. (2018) also referred to the fact that the momentum parameter of BN is essential for fine-tuning and provided a strategy in section A.5. Discussion about this strategy will make this submission more complete.”
>
> A: To clarify, we do not change the default momentum in Batch Normalization in our end-to-end fine-tuning experiments. As the reviewer can appreciate, in principle all the hyper-parameters could be tuned, we choose to focus on the most important ones.
>
> We add a discussion about BN momentum in Appendix D.  Kornblith et al., 2018  found it is critical to decrease the batch normalization momentum parameter from its ImageNet value to max(1 − 10/s, 0.9) where s is the number of steps per epoch. This will change the default BN momentum value (0.9) when s is larger than 100, which means the original dataset size has to be larger than 100*256 = 25.6K. The maximum data size used in our experiments is Caltech-256, which is 15K, so it is not applicable to our experiments.
>
> We do add experiments for exploring the effect of BN momentum by performing similar study as to ELR. We try to identify whether there is an optimal BN momentum for each dataset. For each dataset, we fine-tune the pre-trained model using previously obtained best hyperparameters and only change the BN momentum before fine-tuning. In addition to the default value 0.9, we searched 0.0, 0.95 and 0.99. We believe if BN momentum is critical, we may expect noticeable performance differences. The results are shown in Appendix D. We observe 0.99 slightly improves the performance for some datasets, however, we did not see the significant performance difference among values greater than 0.9. We suspect that when fine-tuning steps is long enough, the dataset statistics will eventually be adapted to the target dataset.
>
> [1] Cui et al. 2018, Large scale fine-grained categorization and domain-specific transfer learning. In CVPR 2018
> [2] Smith et al, Don't decay learning rate, increasing the batch size, ICLR 2018
> [3] Kornblith et al, Do Better ImageNet Models Transfer Better? CVPR 2019

---

> ### Author Response · Authors · 2019-11-15
> **our response 1/2**
>
> Thanks for your detailed and constructive review!  We have made changes and adding additional experiments in the latest revision for answering the following questions.
>
> Q1: “More experiments on other backbones are necessary.”
>
> A: We actually provided experiments with different backbones in Appendix B, including DenseNet-121 and MobileNet-v1. We made consistent observations among the three architectures: 1) the optimal effective learning rate is related with the similarity to source domain. 2) In Appendix C of revised version, we also calculated the EMD based domain similarities with each pre-trained model as the feature extractor and find the relative orders are quite consistent.
>
> Q2: “this submission claims that the regularization methods such as L2-SP may not work on networks with Batch Normalization module. But there is no comparison on networks without BN.”
>
> A: Due to the wide usage of BN layers in recent architectures, especially deep CNNs, it is hard to find pre-trained ImageNet models without BN layers. Instead of verifying the advantage of L2-SP over L2 with pre-trained ImageNet models without BN, we study whether the weight norm changes during training affect the performance. We show though the L2-SP preserves the norm distance with the initial model, there is no difference in the generalization performance. In addition, we also revisited experiments on similar datasets, including Dogs, Caltech and Indoor, we show even with these datasets, L2-SP does not have significant advantage over L2 by preserving the norm similarity. With proper hyperparameter search, L2 has the same effect as L2-SP norm.
>
> Q3: “Providing a complete hyperparameter selecting strategy for fine-tuning could be an important contribution of this submission. I suggest authors to think about it.”
>
> A: Good point. As we discussed in the discussion section, understanding the relationship between the optimal hyperparameters and the domain similarities could be very useful for reducing the number of hyperparameters or search ranges for HPO. We showed the correlation between the domain similarity and the optimal ELR and domain similarity is a good indicator for optimal ELR selection. In Appendix C, we provide a simple hyperparameter selection strategy could be tuning hyperparameter based on the domain similarity. Given a source model, one can calculate its domain similarities and perform exhaustive hyperparameter searches for both similar and dissimilar datasets, and we refer these datasets as reference datasets. Then for the new incoming dataset for fine-tuning, we calculate its domain similarity and compare with the scores of reference datasets, and choose similar ELRs with the closest domain similarity. Note that the optimal ELR is not strictly corresponding to the domain similarity but provides reasonable ranges. We expect more advanced method to be developed as further work.
>
> Q4: “this submission does not propose a proper method for measure the similarity or provide detailed experiments on previous measurements”
>
> A: There are many approaches for measuring the domain similarity. We refer readers to [Cui et al. 2018] for calculating the similarity based on EMD. In Appendix C of the revised version, we introduce the steps for calculating the similarity between two datasets based on EMD. Different with Cui et al, 2018 that uses ResNet101 pre-trained on JFT for feature extraction, which is not publicly available, we propose to use the source model directly as feature extractor for both source and target domains. The distance scale is a bit different from the original paper, but the relative similarities to ImageNet is almost the same, such as Dogs is more similar to ImageNet than Indoor or Cars.
>
> Q5: “It seems that the MITIndoors Dataset is not similar with ImageNet from the semantic view. This submission does not provide similarity measurement between these datasets. Why the optimal momentum is 0? ”
>
> A: In Appendix C of revised version, we calculate the domain similarity with ImageNet for Caltech256 and MIT-Indoor with our ImageNet pre-trained ResNet-101. Their similarities are 0.8916 and 0.8563, which are still larger than Cars (0.8452), Aircrafts (0.8404) and Flowers (0.8444). In Table 2, momentum 0 is better than momentum 0.9 when the learning rate is 0.01 and 0.005, which indicates the ELR with momentum 0.9 is too large and momentum 0 makes the ELR 10x smaller and is more close to the optimal value.

---

### Public Comment · ~Simon_Kornblith1 · 2019-11-08
**Tuning momentum is not important if learning rate is tuned**

While it contains several interesting findings, I do not believe this paper should be accepted as long as it states that "picking the right value for momentum is critical for fine-tuning performance." Section 3.3 of the paper shows that this is false, and what is critical is performing a sufficiently expansive grid search for learning rate. If this is done, there is no need to separately tune momentum. All effects attributed to momentum can be obtained by adjusting learning rate. Both reviewers 2 and 3 recognize this below, but I'd like to provide a signal boost.

---

> ### Author Response · Authors · 2019-11-15
> **our response**
>
> Thanks for the comment, Simon!  Yes, as we stated in section 3.3 “it is effective learning rate that matters for the best performance. It explains why the common practice of changing only learning rate generally works, though changing momentum may result in the same effect”.  We agree that simply emphasize the importance of momentum or any single hyperparameter without context may be miss leading. As we believe learning rate is a “critical” hyperparameter, the aim of the first part is to show momentum is also an equivalent “critical” hyperparameter if other hyperparameter is fixed (e.g., lr=0.01) not well tuned, rather than common belief that 0.9 is the best value or no effect when momentum is tuned.
>
> In fact, as long as the initial learning rate is smaller enough, we can always search for the optimal momentum as momentum is an amplifier, making the ELR larger by a factor of 1/(1-m). Therefore, momentum does determine the search ranges of learning rate. We would like to provide an alternative view, similar as [Smith et al, 2018] where increasing batch size during training has the same effect as decreasing learning rate.
>
> [1] Smith et al, Don’t decay the learning rate, increase the batch size, ICLR 2018

---

### Decision · Program_Chairs · 2019-12-19

**Decision:**

Accept (Poster)

**Comment:**

This paper presents a guide for setting hyperparameters when fine-tuning from one domain to another. This is an important problem as many practical deep learning applications repurpose an existing model to a new setting through fine-tuning.  All reviewers were positive saying that this work provides new experimental insights, especially related to setting momentum parameters. Though other works may have previously discussed the effect of momentum during fine-tuning, this work presented new experiments which contributes to the overall understanding. Reviewer 3 had some concern about the generalization of the findings to other backbone architectures, but this concern was resolved during the discussion phase. The authors have provided detailed clarifications during the rebuttal and we encourage them to incorporate any remaining discussion or any new clarifications into the final draft.